# Mapping cannabis potency in medical and recreational programs in the United States

**Mary Catherine Cash**[1☯], **Katharine Cunnane**[2☯], **Chuyin Fan**[1], **E. Alfonso Romero-Sandoval**[ID][2]*

**1** The University of North Carolina Eshelman School of Pharmacy, Chapel Hill, NC, United States of America,
**2** Department of Anesthesiology, Wake Forest University School of Medicine, Winston-Salem, NC, United States of America

☯ These authors contributed equally to this work.
* earomero.sandoval@gmail.com

**Data Availability Statement:** All relevant data are within the paper and its Supporting Information files.

**Funding:** Funding from Department of Anesthesiology at Wake Forest University School

## Abstract

Cannabis related online searches are associated with positive attitudes toward medical cannabis, particularly when information is obtained from dispensaries. Since pain is the main reason for medicinal cannabis use, information from dispensary websites has the potential to shape the attitude of pain patients towards cannabis. This is relevant because cannabis has demonstrated efficacy in neuropathic pain with low tetrahydrocannabinol (THC) concentrations (< 5–10%), in contrast to potent cannabis (>15% THC), which is highly rewarded in the recreational realm. The role of CBD in pain is not clear, however it has gained popularity. Thus, we hypothesize that the potency of medical cannabis that is advertised online is similar to the cannabis advertised for recreational purposes, which would potentially create a misconception towards medical cannabis. The current lack of knowledge surrounding advertised potencies in the legal cannabis market limits the ability to generate clear policies regarding online advertising to protect patients that are willing to use cannabis for their condition. Thus, we evaluated the advertised THC and CBD content of cannabis products offered online in dispensaries in the United States to determine products' suitability to medicinal use and compare the strength of products offered in legal medical and recreational programs. We recorded THC and CBD concentrations for all herb cannabis products provided by dispensary websites and compared them between or within states. Four Western states (CA, CO, NM, WA) and five Northeastern states (ME, MA, NH, RI, VT) were included. A total of 8,505 cannabis products across 653 dispensaries were sampled. Despite the clear differences between medicinal and recreational uses of cannabis, the average THC concentration advertised online in medicinal programs was similar (19.2% ±6.2) to recreational programs (21.5% ±6.0) when compared between states with different programs, or between medicinal and recreational programs within the same states (CO or WA). Lower CBD concentrations accompanied higher THC products. The majority of products, regardless of medicinal or recreational programs, were advertised to have >15% THC (70.3% - 91.4% of products). These stated concentrations seem unsuitable for medicinal purposes, particularly for patients with chronic neuropathic pain. Therefore, this information could induce the misconception that high potency cannabis is safe to treat pain. This data is consistent with reports in which THC and CBD in products from legal dispensaries or in

of Medicine (E.A.R.s.). The funders had no role in study design, data collection and analysis, decision to publish, or preparation of the manuscript.

**Competing interests:** The authors have declared that no competing interests exist.

nationwide products from the illegal market were actually measured, which indicates that patients consuming these products may be at risk of acute intoxication or long-term side effects. Our study offers grounds to develop policies that help prevent misconceptions toward cannabis and reduce risks in pain patients.

## Introduction

The practice of pain management has come under scrutiny in recent years with the rise of the opioid epidemic in the United States (U.S.). Physicians continue to search for alternatives when opioids, anticonvulsants, or antidepressants provide no relief or result in adverse effects. Cannabis offers an alternative to pain management. While it is challenging to determine a cause and effect relationship between the implementation of cannabis legislation and a decline in opioid prescribing practices given the concurrent rise of legislation aimed at restricting and monitoring opioid prescribing, states with legalized medical cannabis programs have witnessed a decline in the number of opioid prescriptions and prescribed doses, particularly among younger cohorts and for schedule III opioids[1–4]. Interestingly, recreational cannabis legislation has not demonstrated a significant impact on opioid prescribing in privately insured adults [5].

Pain is the foremost reason patients visit cannabis dispensaries across the U.S. [1, 6] As of October 2019, 39 states and Washington D.C. have legalized cannabis for medical use in the U. S. Of those, 10 states and Washington D.C. have also legalized cannabis for recreational use. Thirty-four states list pain as a qualifying condition and six have approved only the use of CBD oils for medical purposes. Thus, the U.S. represents a largely populated geographical area in which cannabis is becoming rapidly legal and accessible in a non-uniformly regulated market in contrast to other countries such as Uruguay, which has slowly implemented its cannabis policies in order to address challenges and insure the safe and successful implementation of a nation-wide cannabis program. This tendency towards rapid cannabis legalization in the U.S. has been accompanied with robust dissemination of information using new technologies, namely online advertisements. In fact, marijuana or cannabis online searches have grown exponentially during the last decade across the U.S. [7]. Interestingly, online presence of cannabis products is associated with positive attitudes towards the medicinal properties of cannabis [8]. More importantly, it has been demonstrated that information provided by dispensaries is highly regarded by patients as safe and reliable [9]. Providing wrong information via online advertisements represents a high risk for public health, as evidenced by the recent concern expressed by the Federal Drug Administration (FDA) towards online offers and claims about cannabis products [10]. Therefore, understanding what information is provided online by dispensaries will provide a better understanding on how this shapes the attitude of pain patients towards cannabis products for medical purposes.

Inhaled cannabis has been proven effective for the treatment of various types of chronic pain ranging from neuropathic pain to diabetic nephropathy and has a more favorable pharmacokinetic profile than oral formulations [11–15]. Thus, knowing what type of cannabis herbal products are offered online is relevant for pain patients. The main components found in cannabis are tetrahydrocannabinol (THC), the constituent of cannabis responsible for inducing intoxicating effects, and cannabidiol (CBD), which is devoid of intoxicating effects. THC is the primary constituent of cannabis and is responsible for its analgesic and euphoric effects, as well as its adverse effects. Even though the analgesic effects of THC are dose

dependent [16, 17], an increase in THC content (%THC for inhaled formulations) increases the frequency of adverse events [11, 14, 15], and might increase pain, as demonstrated in experimental pain studies in humans [17, 18]. An array of studies have demonstrated efficacy in pain reduction with minimal and tolerable intoxicating effects with THC concentrations lower than 5% or 10% [14, 15, 19–22]. Alternatively, CBD has demonstrated the potential to ameliorate the psychotic- and anxiety-inducing effects of THC [23, 24] and may mitigate its memory-impairing effects as well [25, 26]. Thus, the concentration of THC and CBD seems to contribute to the overall psychotropic and therapeutic effects patients experience. Even though a recent article has described some aspects of U.S. dispensaries' online practices and information about cannabis [27], thus far there are no reports on the strength of cannabis products offered in legal medical cannabis programs across the United States (U.S.), and how this compares to recreational programs. The clinical relevance of such information relies on the fact that this information could be considered safe by patients, which may strongly influence their attitude towards cannabis as medicine [9]. In addition, this study seeks to uncover whether the reported potency of cannabis products marketed online are in line with the few reports on the potency of products measured at different times, settings, techniques, and laboratories [28, 29]. This type of information is relevant from a policy or regulatory standpoint as it would allow us to evaluate the accuracy of advertised cannabis potency. Altogether, this information could guide more immediate decisions by regulatory agencies to request changes in online product offers from dispensaries just as the FDA has done recently through warning letters directed to remove online unsubstantiated claims of cannabis products [10].

Due to the continuous historic increase in THC content in cannabis products (>15%) on the illegal market in the U.S. [30–32], we hypothesize that the THC content of cannabis offered in legal medical programs are higher than the ideal concentration required for the treatment of pain (<5–10%) and that the THC and CBD content of products found in medicinal dispensaries are comparable to those found in recreational programs. This study aims to map the THC and CBD content available on dispensary websites in states with legalized medical and/or recreational cannabis programs in the Northeastern and Western regions of the U.S. in order to characterize the variety of products available to patients across the country. This study will also evaluate the appropriateness of available products for medicinal use based on the advertised potency and determine whether a difference exists between the strength of cannabis offered by medical and recreational programs. Our study has the potential to impact public health as it provides the foundation to develop and implement evidence-based policies and regulations for online herbal cannabis advertisement for medical purposes, which in turn will result in more realistic patient attitudes towards cannabis.

## Materials and methods

### Inclusion and exclusion criteria

States with legalized medical and/or recreational cannabis programs were identified. The number of licensed dispensaries in each state was determined. Dispensaries were evaluated for online presence. Those with established websites (not including a profile on Leafly or Weed-Maps) were assessed for the availability of THC and CBD data. Only states that have legalized cannabis for the treatment of pain management were included. States or programs who do not allow for the inhaled administration of cannabis were excluded, as were those recreational cannabis programs that have yet to take effect despite being legalized (as was the case with Massachusetts and Maine). States were considered to have an "active" program if there were licensed dispensaries open and operating. Two distinct geographical locations containing a group of states meeting the above criteria were identified—the Northeast region and the Western region

of the United States. Selected states from the Northeast region include Maine (ME), Massachusetts (MA), New Hampshire (NH), Rhode Island (RI), and Vermont (VT). Selected states from the Western region include Colorado (CO), New Mexico (NM), Washington (WA), and California (CA). Of the selected states, all Northeastern states and NM had legalized cannabis only for medicinal use. CO, WA and CA have legalized cannabis for recreational and medicinal use. Due to the variation in geographical size and cannabis program size between the Northeast and the West, each region had unique set of inclusion and exclusion criteria and a unique protocol for dispensary sampling. More information on licensed dispensaries, dispensaries sampled, and their inclusion criteria could be found in Supporting Information (S1 and S2 Tables).

## Northeast

States were considered for inclusion if more than 50% of its dispensaries had an online presence. In addition, states were only included if more than 50% of dispensaries provided THC content on their websites. A dispensary was deemed to have THC content available online if more than 50% of the products available at that dispensary had THC content (%) listed on their website. The size of the medical and recreational cannabis programs in the Northeast made it possible to sample every licensed dispensary in each state for online presence and availability of THC content. A list of licensed dispensaries in each state was obtained from the Maine Department of Health and Human Services (N = 8), the Massachusetts Medical Use of Marijuana website (N = 20), the New Hampshire Department of Health and Human Services (N = 4), the Rhode Island Department of Health (N = 3), and the Vermont Department of Public Safety Marijuana Registry (N = 5). Data for the Northeastern states was collected between March and May of 2018.

## West

Due to the size of the medical and recreational cannabis programs in the West a representative portion of dispensaries was sampled in states with more than 100 dispensaries. The process by which dispensaries were selected varied from state to state depending on the availability of licensed dispensary data on government websites. Sampled dispensaries were geographically dispersed throughout each sampled state in order to maintain a representative sample. If a state had more than 100 licensed dispensaries, at least 30% were sampled for online presence and THC content. Western states were considered for inclusion if more than 50% of the sampled dispensaries had an online presence. The criteria for Northeastern states to have more than 50% of the dispensaries include THC content online was not considered when determining the inclusion of Western states as many Western dispensaries elect not to provide cannabinoid content on their websites. Similar to the Northeastern states, dispensaries were determined to have available THC content if more than 50% of products listed on a dispensary website reported THC content.

Sampling of dispensaries in the West varied by state. Dispensaries in CO were identified from a list of licensed dispensaries provided by the Colorado Department of Revenue. At least one dispensary in every CO city with an open and licensed dispensary was sampled (N = 200). In New Mexico, all 72 dispensaries listed by the New Mexico Department of Health were sampled. The Washington State Liquor and Cannabis Board classifies each dispensary as a retail or medical dispensary. One retail and one medical dispensary from each county in WA was selected utilizing an interactive map provided by the Washington State Liquor and Cannabis Board website (N = 80). At the time of data collection, the California Bureau of Cannabis Control did not provide information regarding the names and locations of licensed dispensaries in

the state. Therefore, the state of CA was divided into 3 geographical regions: Northern CA, Mid-CA and Southern CA. Dispensaries were sampled from each region using Google Maps, insuring even geographical distribution of dispensaries (N = 268). After the California Bureau of Cannabis Control released names of licensed dispensaries, we confirmed that all 26 California dispensaries with online presence and available THC data included in our analysis were indeed licensed by the state of California at the time of data collection.

### Data collection

Cannabinoid data was collected for all chemotypes of THC-rich, CBD-rich, and mixed THC/CBD flowers and pre-rolls. It is worth mentioning that the dispensary websites referred to the advertised products as "strains" and classified such products as "Sativa", "Indica" or "Hybrid". For products that provided a range of THC or CBD, an average was calculated (i.e. THC 15–17%, a THC of 16% was recorded). For THC or CBD reported as being less than or equal to a particular value, that number was reported as the THC or CBD value (i.e. CBD <0.05%, a CBD of 0.05% was recorded).

### Statistical analysis

Mean and standard deviation were determined for each state. Histograms containing THC and CBD concentrations were constructed and three concentration ranges were identified: THC <5%, THC 5–10%, and THC 10–15%, and THC > 15%. Student's T test or One-way ANOVA and Turkey's multiple comparison test were used. A p<0.05 was considered statistically significant. All data are presented as (mean ± SD; median 25% percentile, 75% percentile).

### Results

We first quantified the THC and CBD content of all herb cannabis products offered in the surveyed dispensaries from states with legal medicinal marijuana programs and compared it with similar products offered in dispensaries from states with legal recreational/medicinal marijuana programs. All data per product and studied states are available in Supporting Information. The average THC concentration in products from medicinal programs was significantly lower (19.3% THC ± 0.17; 20% THC 16.8, 22.9) than those found in products from recreational programs (21.5% THC ± 0.068; 21.2% THC 18.8, 23.9 P<0.0001, Fig 1A). The average CBD concentration in products from medicinal programs was also significantly higher (2.0% CBD ± 4.5; 0.2% CBD 0, 0.6) than those found in products from recreational programs (1.3% CBD ± 3.6; 0.2% CBD 0, 0.6 P<0.003, Fig 1B). However, these average values are not representative of the wide range in the concentrations of THC and CBD. For example, products ranged from 0 to 35% THC in medical programs, while THC concentrations ranged from 0 to 45% in recreational programs. Most products in medical programs contain less than 5% CBD with some containing 15%, while most products in recreational programs contain between 0 to 18% CBD, with some above 20% and a few products above 40%.

We then sought to determine whether this variability persists in a state-by-state basis. To this end, we performed a similar quantification of the THC and CBD content of all herb cannabis products by state and conducted comparisons between all states with legal medicinal and recreational/medicinal marijuana programs. We found that average THC concentrations were numerically similar in all surveyed states (descriptive statistics could be found in S3 Table), but some statistical differences were found (ranging from 15.2% THC average [17.4%, median] in VT to 21.72% THC average [21.3%, median] in WA, Fig 2A, S4 Table). Frequency histograms reveal that most products in all surveyed states contain THC concentrations between 15% and

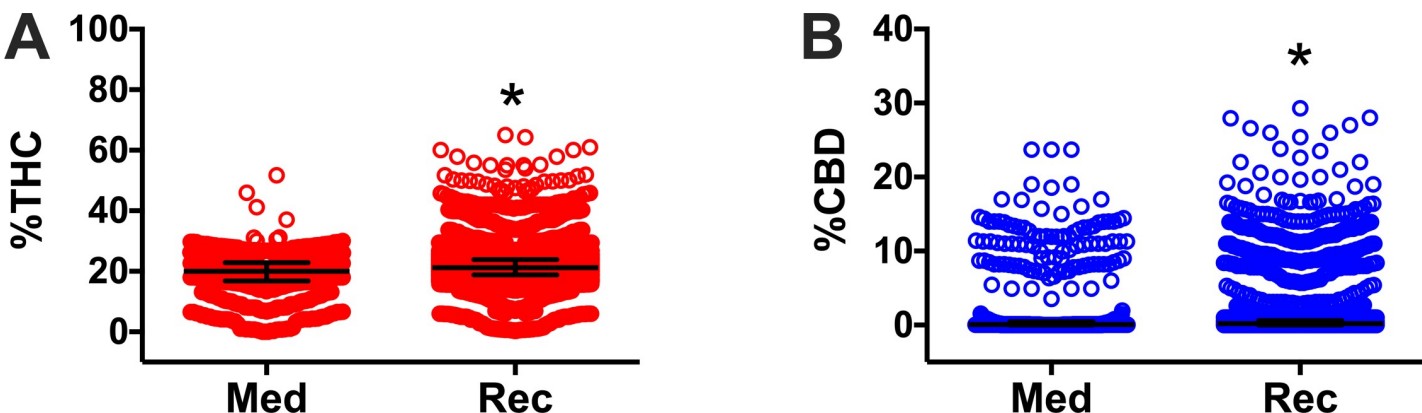

**Fig 1.** Percent THC (A) and percent CBD (B) in legalized medicinal or recreational dispensaries in the Northeastern and Western regions of the United States. Herb cannabis products offered in dispensaries from states with medicinal only (Med; ME, MA, NH, RI, VT, and NM) and recreational and medicinal (Rec; CA, CO, and CA) programs were included for analysis and represented individually as circles. Data are presented as mean ± SD. *P<0.05 vs. medicinal, by Welch's t test. The outlier products in the graph might represent herbal products infused with concentrated THC or CBD extracts, however we did not observe such a claim in the studied websites where these products were offered.

30% (Fig 2B and 2C). Individual (per state) frequency histograms in relation to THC content products could be found in Supportive Information (S1 Fig). Additionally, individual histograms to depict individual products with their respective THC content (potency) could be found in Supportive Information (S2 Fig). Comparisons between THC concentrations offered in different states may be found in S4 Table. When CBD was analyzed, we found that average CBD concentrations were more variable between states (ranging from 0.9% CBD average [0.4%, median] in ME to 8.3% CBD average [8.3%, median] in VT, Fig 2D, S3 Table). Frequency histograms reveal that most products in all surveyed states contain CBD concentrations between 0% and 2% (Fig 2E and 2F). Individual (per state) frequency histograms in relation to CBD content products could be found in Supportive Information (S3 Fig). Additionally, individual histograms to depict individual products with their respective CBD content (potency) could be found in Supportive Information (S4 Fig). Comparisons between CBD concentrations offered in different states may be found in S5 Table.

The large variability in THC and CBD concentration remains when the analysis was conducted in each studied state, which could result in a misleading hypothesis evaluation. Therefore, we conducted a more detailed analysis based on THC concentrations that represent the clinical or recreational adequacy of the products available in medicinal and/or recreational programs in the studied states. Thus, we divided herb products into 4 categories based on the level of THC: <5% THC, ≥5≤10% THC, >10≤15% THC and >15%THC. We observed in all states that the majority of THC products had >15% THC (ranging from 70.3% of products in ME to 91.4% of products in CO). Excluding VT and NH, the second most abundant category in all states was >10≤15% THC. In all states but VT and NH, the third most abundant category was ≥5≤10% THC. The least abundant category in all states except for NH was <5% THC. In VT, the second most abundant category was ≥5≤10% THC followed by >10≤15% THC. In New Hampshire, the second most abundant category was ≥5≤10% THC closely followed by the <5% THC category, with only 3.77% of products falling in the >10≤15% THC range. Percentage of products in each THC category may be found in Fig 3.

To further compare the potential variability in product THC concentrations in states with legal medicinal and recreational/medicinal marijuana programs, we plotted the concentration of THC in individual herb products in each THC category separated by state (Fig 4, S6 Table). The most salient findings are that ME and VT offered no products in the <5% THC category,

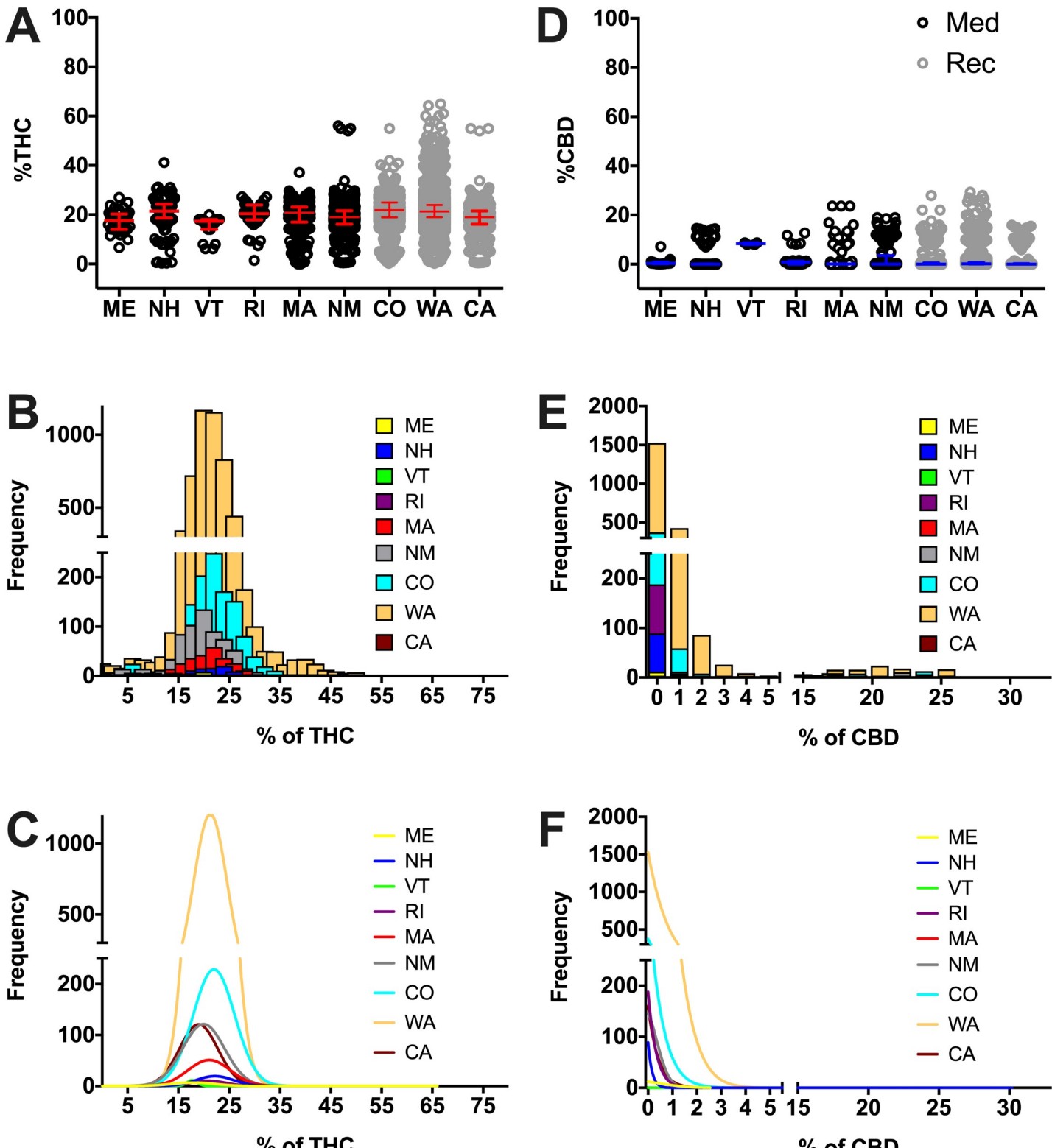

**Fig 2.** Percent THC (A) and percent CBD (D) per state. Percent THC and CBD was obtained from dispensary websites for herb cannabis products in the surveyed states and plotted as circles. States with medicinal only (Med; black circles) and recreational and medicinal (Rec; grey circles) programs were compared. Data are presented as median with interquartile range. For clarity we do not show the statistic differences among states, statistical analysis results and descriptive statistics could be found in S4 and S5 Tables. Frequency histograms of products in relation to THC (B) or CBD (E) and their respective Gaussian fit curves (C and F) show the

abundance of products of given potencies. The outlier products in the graph might represent herbal products infused with concentrated THC or CBD extracts, however we did not observe such a claim in the studied websites where these products were offered.

and RI only offered 1. These are states with only medicinal programs. All states ranged their THC levels similarly in the other THC categories. However, in the >15% THC category, CO (ranging from 15.1% THC to 55% THC), WA (ranging from 15.10% THC to 65.00% THC), and CA (ranging from 15.01% THC to 55.00% THC) offered cannabis products that spanned a much wider range of THC concentrations than the other surveyed states. Significant difference among states in the >15% THC category can be found in S7 Table.

In order to analyze the ratio of THC to CBD, the concentration of CBD in each product was separated by state and plotted within its THC category (Fig 5, S8 Table). In the <5% THC category, there were no significant differences in average CBD concentration between states. Interestingly, most of the Northeastern states (medical programs) have the smallest range of

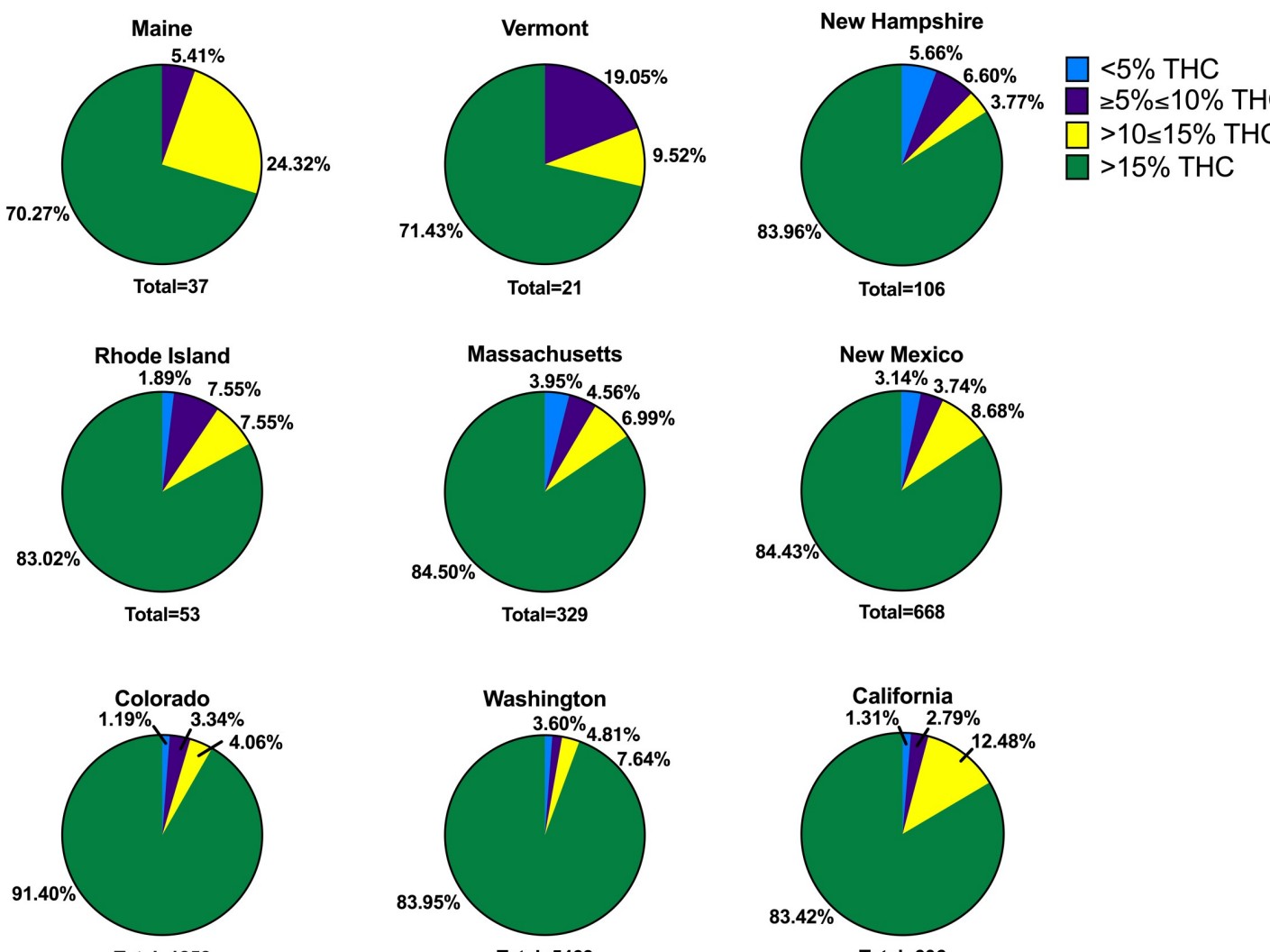

**Fig 3. Proportion of products with different levels of THC per state.** Products are presented in the following categories: <5% THC (blue), ≥5%≤10% THC (purple), >10≤15% THC (yellow) and >15% THC (green) in all surveyed states. The total number of products analyzed per state is presented at the bottom of every graph (state) with medicinal programs (ME, VT, NH, RI, MA, and NM) or medicinal and recreational programs (CO, WA, and CA).

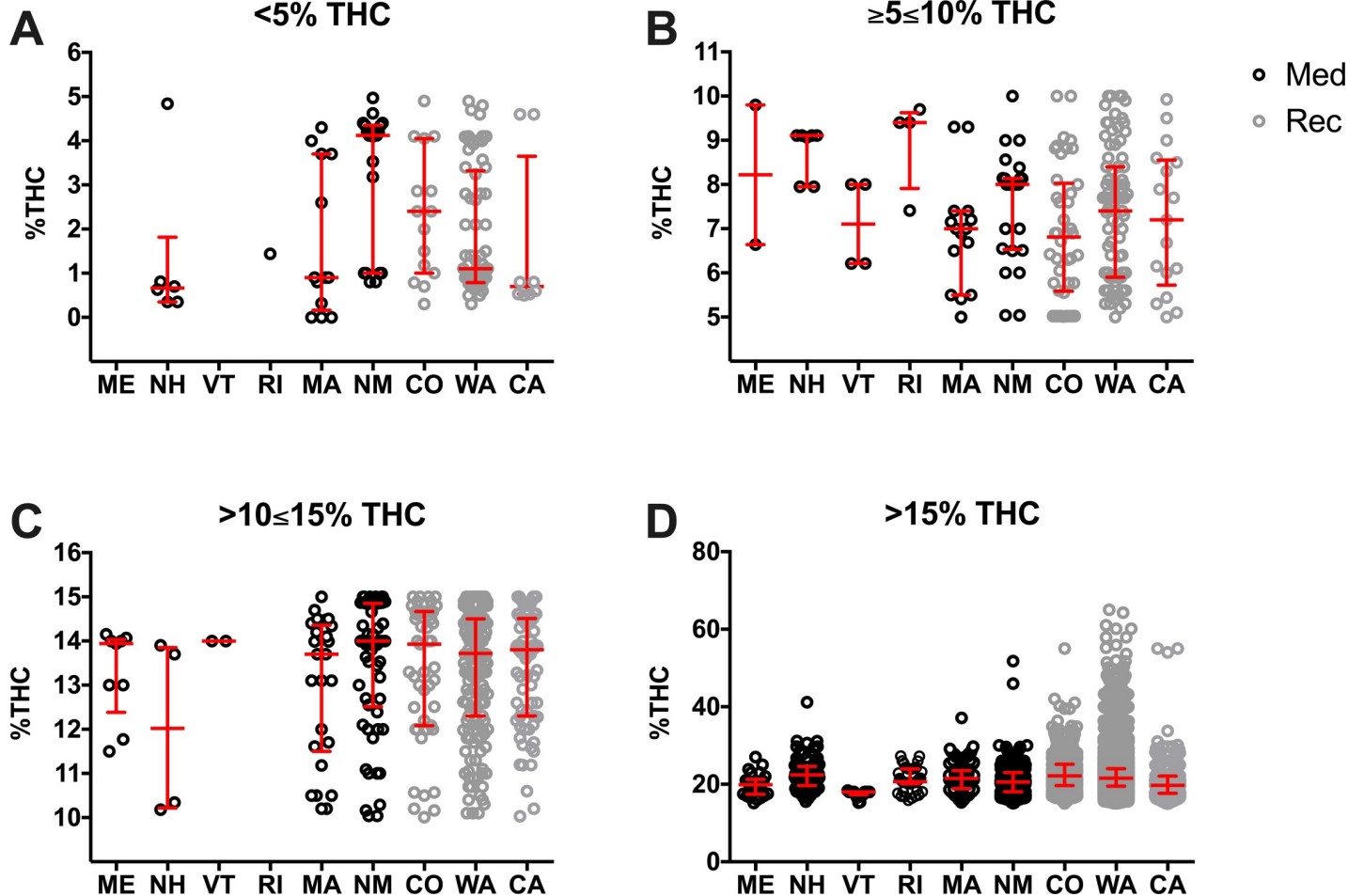

**Fig 4. Percent THC at different strength product categories based on THC content per state.** Products are presented in the following categories: <5% THC (A), ≥5≤10% THC (B), >10≤15% THC (C), and >15% THC (D). Every herb cannabis product in the surveyed states is plotted as an open circle. States with medicinal only (Med; black circles) and recreational and medicinal (Rec; grey circles) programs were compared. Data are presented as median with interquartile range. Statistic significant changes (P<0.05) were observed between the following states: NM vs. WA in <5% THC; NH vs. CO in ≥5≤10%. One-way ANOVA and Tukey's multiple comparison test. For clarity we do not show the statistic differences among states in the >15% category, these statistical analysis results and descriptive statistics could be found in S6 and S7 Tables. The outlier products in the graph might represent herbal products infused with concentrated THC or CBD extracts, however we did not observe such a claim in the studied websites where these products were offered.

CBD content in these categories. However, WA and MA offered products with a much wider range of CBD concentrations (approximately ranging from 0% to 25% CBD) in the <5% THC category. It is noteworthy that MA, NM, CO, WA, and CA offered a variety of different CBD concentrations between 0% CBD and 30% CBD in the ≥5≤10% THC. In the >10≤15% and >15% THC categories, the levels of CBD were much lower than in the other THC categories.

Next, we selected the two states where medical and recreational cannabis products are sold separately. In CO, the majority of dispensaries offered separate medical and recreational menus within the same establishment. In Washington, most medical dispensaries existed as entirely separate entities from recreational dispensaries. Thus, we were able to draw comparisons between recreational and medical products of cannabis within the same state in two different scenarios.

The average THC concentrations were numerically similar in all CO and WA programs (approximately 21% average THC [21–22 median], Fig 6, S9 Table). Both medical and

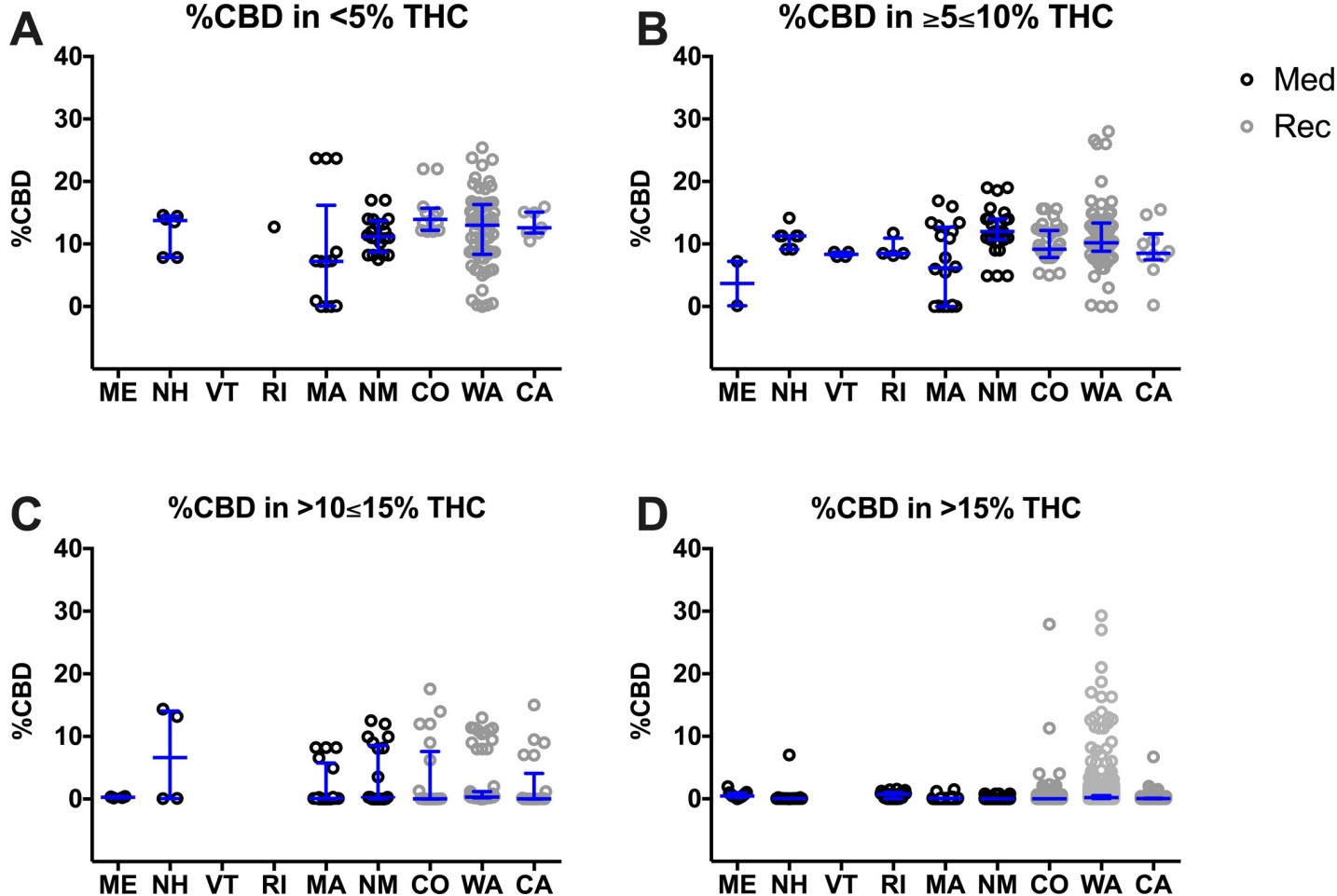

**Fig 5. Percent CBD at different strength product categories based on THC content per state.** Products are presented in the following categories: <5% THC (A), ≥5≤10% THC (B), >10≤15% THC (C), and >15% THC (D). Every herb cannabis product in the surveyed states is plotted as an open circle. States with medicinal only (Med; black circles) and recreational and medicinal (Rec; grey circles) programs were compared. Data are presented as median with interquartile range. Statistic significant changes (P<0.05) were observed between the following states: MA vs. WA and MA vs. NM in ≥5≤10%; MA vs WA, CO vs. WA, NM vs. WA in >15%. One-way ANOVA and Tukey's multiple comparison test. Descriptive statistics could be found in S8 Table.

recreational dispensaries in WA offered a wider variety of products, with some products containing above 40% THC. The average CBD concentrations were numerically similar in all CO and WA programs (approximately 1% average CBD [0 median], Fig 6, S9 Table). Medical and recreational dispensaries in both states offered a wider variety of products, with some products containing nearly 30% CBD.

Next, we divided herb products into the same four %THC categories as before (Fig 7). We observed in both medicinal and recreational programs, the majority of THC products had >15% THC (ranging from 89.4% of CO medical products to 95.7% of WA medical products). The second most abundant THC category in all conditions was >10≤15% THC. The third most abundant category in all conditions excluding WA medical was ≥5≤10% THC. In all states except for WA medical, the least abundant category was <5% THC. In WA medical products, the third most abundant category was <5% THC, making ≥5≤10% THC the least abundant category.

To further compare the distribution of THC concentrations of products in states with separate legal medicinal and recreational programs, the concentration of THC in individual

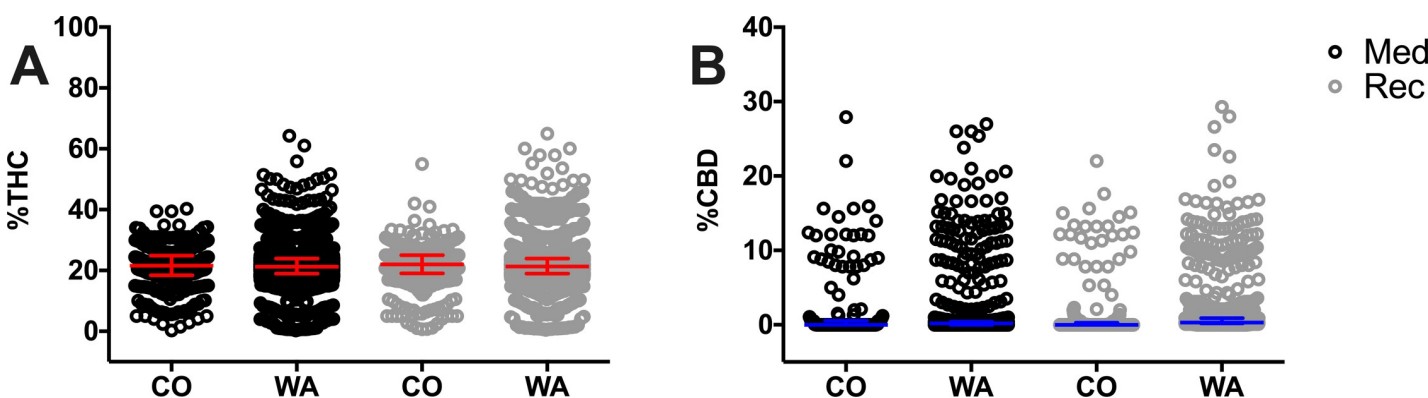

**Fig 6.** Percent of THC (A) and percent CBD (B) in CO and WA medicinal and recreational programs. Data for CO was obtained from the respective medicinal or recreational menus offered in the same dispensary, while data for WA was obtained from either medicinal dispensaries or recreational dispensaries, which exist as independent entities. Herb cannabis products and their THC and CBD percent were included for analysis and represented individually as circles. Data are presented as median with interquartile range. No statistical differences were found among states. Descriptive statistics could be found in S9 Table. The outlier products in the graph might represent herbal products infused with concentrated THC or CBD extracts, however we did not observe such a claim in the studied websites where these products were offered.

cannabis products was plotted in each THC category, separated by state and program (Fig 8). We observed similar averages and ranges of THC concentrations in all conditions in <5%, ≥5≤10%, and >10≤15% THC categories (S10 Table). In the >15% THC category, WA shows a wider range of THC concentrations, but recreational products (22.76% THC ± 5.5; 21.6% THC 19.6, 24.1) were significantly higher in THC than WA medicinal products (22.2% THC ± 4.6; 21.5% THC 19.4, 24, P = 0.0001).

The concentration of CBD in each individual herb product was again separated by program and plotted within its THC category (Fig 9). Average CBD values were similar within each THC division (S11 Table), but WA displayed a wider range of CBD concentrations. In the >15% THC category, average CBD concentrations were numerically similar and low (<0.7%). However, there were a number of products with elevated CBD as well as elevated THC. CBD concentrations got as high as 27–30% in CO medical, WA medical, and WA recreational products. Products in CO recreational products only got as high as 10–12% CBD.

## Discussion

The first major observation of our study was that the average concentration of THC in all states was two to three times the THC content known to be efficacious in the treatment of pain (i.e. >5–10%). The second major finding of our study was that a vast majority of products in all states, including medical-only programs, contained THC designed for recreational use (i.e. > 15%). Patients who find this information in their online searches may subsequently deem high potency products suitable for medical purposes, placing themselves at higher risk of cannabis intoxication. Severe intoxication, hyperemesis, psychiatric symptoms, and severe cardiovascular events have been reported to be a major cause of cannabis-related visits to emergency departments in Colorado [33]. Such undesirable adverse effects may lead to a perception of treatment failure in patients who have already failed traditional pain management therapies, while dependence may potentiate the long-term use of high potency cannabis. The prolonged use of high potency cannabis increases the risk for psychotic disorders by 5-fold in daily users compared to never users [34], increases the risk of memory impairment and paranoia [35–37], and is associated with cannabis admissions to drug treatment [38]. People who use cannabis for medicinal purposes have more frequently reported daily or almost daily cannabis use,

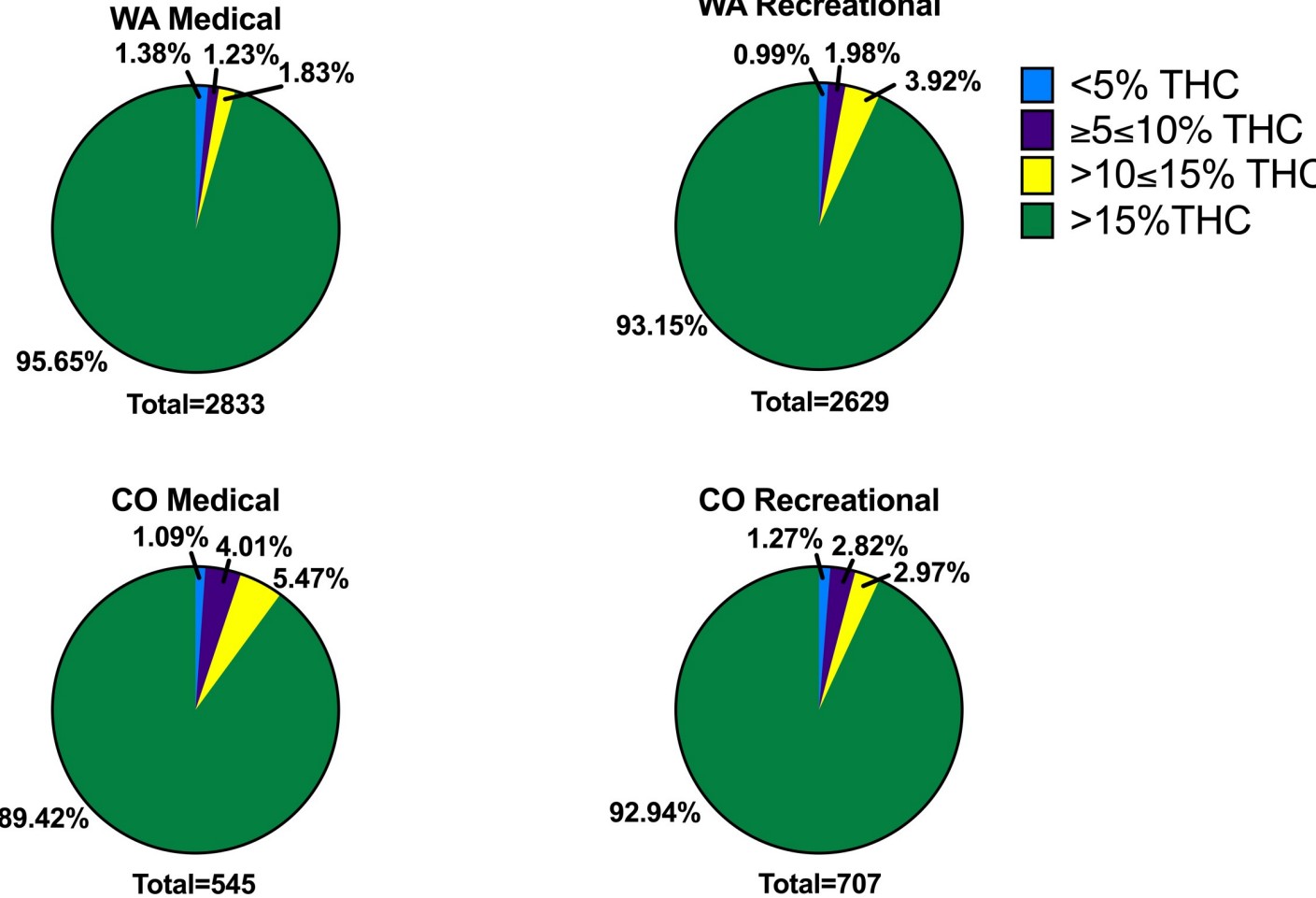

**Fig 7. Proportion of products with different levels of THC in CO and WA medicinal and recreational programs.** Products are presented in the following categories: <5% THC (blue), ≥5%≤10% THC (purple), >10≤15% THC (yellow) and >15% THC (green) in WA medicinal or recreational dispensaries (top panel, WA Medical and WA Recreational), and CO medicinal or recreational menus (bottom panel, CO Medical and CO Recreational). The total number of products analyzed per program is presented at the bottom of every graph (state and program).

suggesting the need for the provision of chemotypes that optimize pain management while limiting adverse effects (i.e. low THC) [39].

Our findings demonstrate that medicinal programs are providing chemotypes that are comparable in potency to those offered by recreational dispensaries. This lack of difference was also apparent when comparing medicinal and recreational programs within the same region. No difference in THC concentrations was observed between medicinal and recreational programs in the same state, neither when medicinal and recreational programs exist in different dispensaries (WA) nor when they exist in the same dispensary in different menus (CO). These findings suggest that, no matter how compared, there appears to be no clinically meaningful difference between medicinal and recreational cannabis potencies across the country.

We did not map concentrations in all dispensaries in all U.S. states where cannabis is legalized for either medical treatment of pain or recreational use; however, our study covers the two regions where most states have such programs. In contrast to smaller yet well-designed studies, our study covers Western and Northeastern regions from the U.S. and 8,505 cannabis

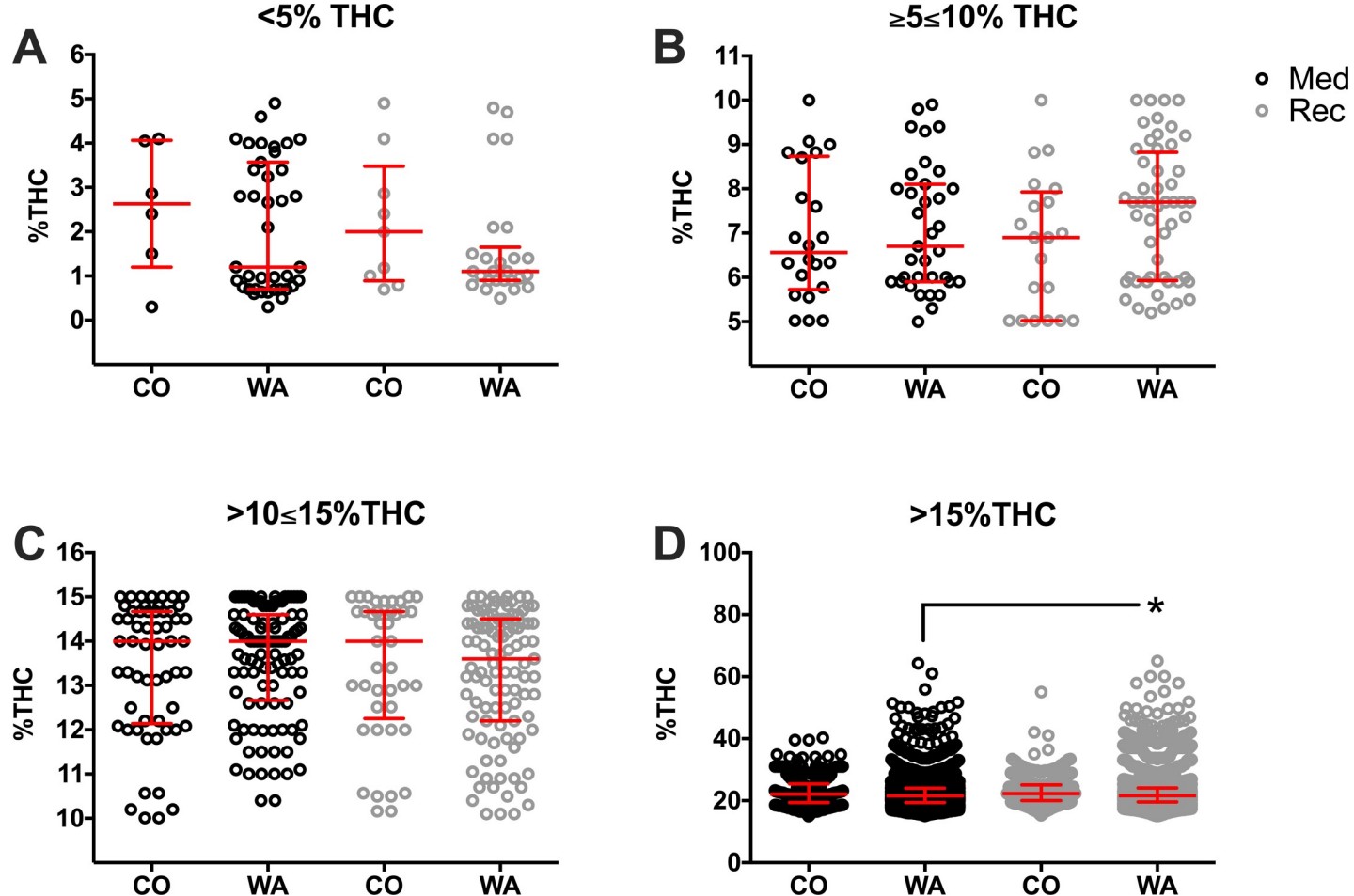

**Fig 8. Percent THC at different strength product categories based on THC content in CO and WA medicinal and recreational programs.** Products are presented in the following categories: <5% THC (A), ≥5≤10% THC (B), >10≤15% THC (C), and >15% THC (D) in CO recreational or medicinal menus and WA recreational or medicinal dispensaries. Every herb cannabis product in the surveyed states and programs are plotted as open circles. Medicinal programs (Med; black circles) and recreational programs (Rec; grey circles) were compared. Data are presented as median with interquartile range. *P<0.05 Recreational WA vs. Medicinal WA (linked line). One-way ANOVA and Tukey's multiple comparison test. Descriptive statistics could be found in S10 Table. The outlier products in the graph might represent herbal products infused with concentrated THC or CBD extracts, however we did not observe such a claim in the studied websites where these products were offered.

chemotypes/products across 653 dispensaries [27]. Therefore, our results are likely a fair representation of the potency of online products offered in dispensaries across the U.S. It is challenging to generalize our findings to states not included in our study given the difference in legal requirements, size of state, population, and other demographics may drive the production and dispensing patterns in each state. Furthermore, state legal requirements differ in regard to home cultivation of cannabis, a population that was not represented in this study yet may also drive demand and further influence the production and dispensing in each state. However, our findings appear to be consistent and surprisingly uniform within and between the Western and Northeastern regions, which comprise very diverse geographic and demographic size, and differ widely in their cannabis program legal requirements. Our study indicates that both recreational and medicinal programs in various and diverse regions of the country are contributing to and reflecting the national trend towards increasing potency of cannabis.

While few dispensaries indicate how their reported THC concentrations were measured or estimated, reported concentrations appear to be comparable to those measured in common

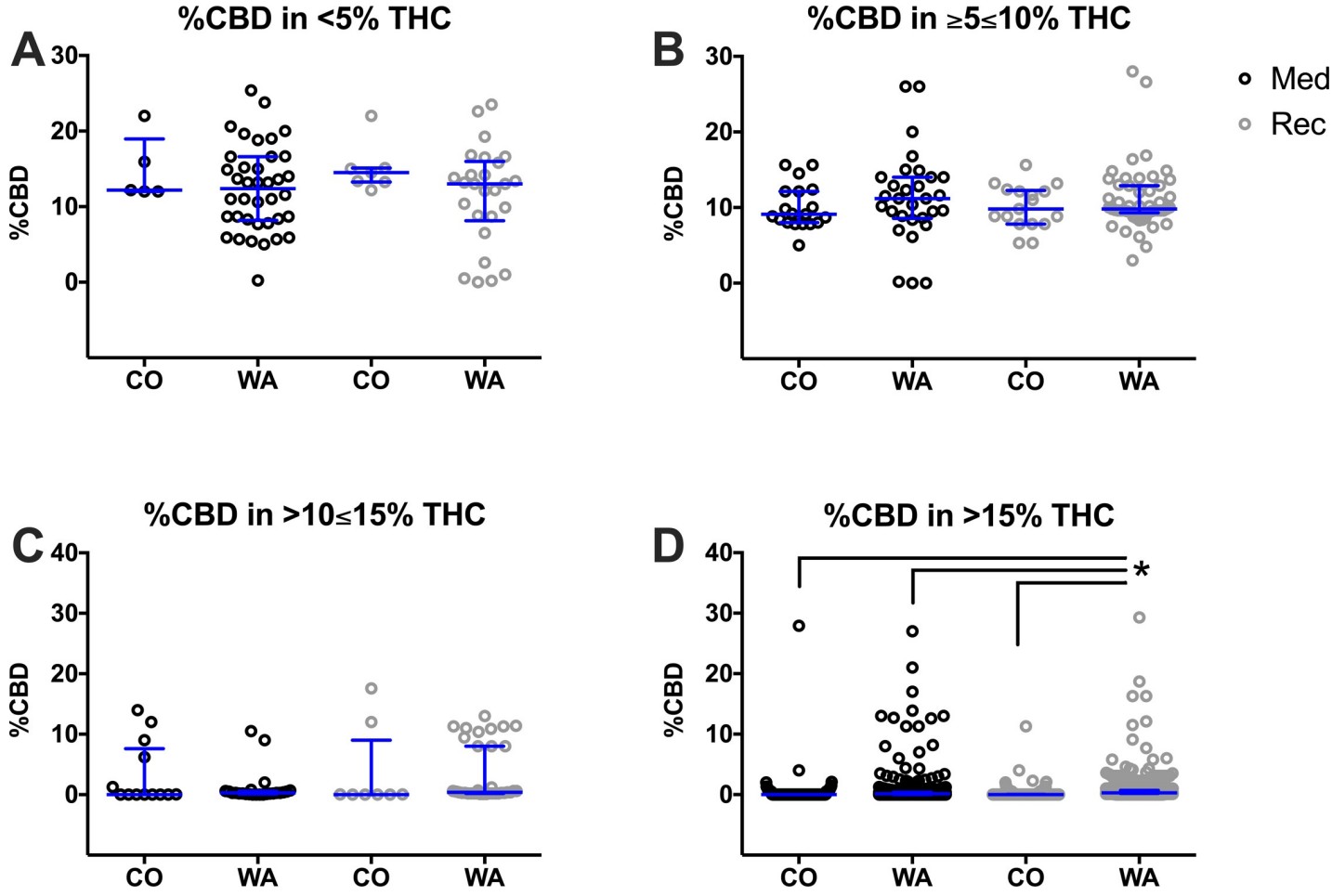

**Fig 9. Percent CBD at different strength product categories based on THC content in CO and WA medicinal and recreational programs.** Products are presented in the following categories: <5% THC (A), ≥5≤10% THC (B), >10≤15% THC (C), and >15% THC (D) in CO recreational or medicinal menus and WA recreational or medicinal dispensaries. Every herb cannabis product in the surveyed states and programs are plotted as open circles. Medicinal programs (Med; black circles) and recreational programs (Rec; grey circles) were compared. Data are presented as median with interquartile range. *P<0.05 Recreational WA vs. Medicinal WA, Medicinal CO, and Recreational CO (linked line). One-way ANOVA and Tukey's multiple comparison test. Descriptive statistics could be found in S11 Table.

products from dispensaries located in CO, CA, and WA. The average THC concentration in these products were consistently above 15% [40]. Similarly, our data are in line with THC and CBD tested in samples seized by the DEA in 2015 [41] and 2017 [30, 42], with an average THC of 20% and 17.8% respectively. The results of our study appear to be in agreement with such results, assuming that the trend toward higher concentrations of THC and predominance of high potency cannabis chemotypes continued just as it has since the 1980s [20, 31]. Average CBD content of products seized in 2017 was 0.15%, compared to 0.41% in 2008, resulting in an increase in the THC/CBD ratio from 23 in 2008 to 104 in 2017 [30, 42]. Our study identified a greater number of products with high THC content compared to fewer products with some CBD content. This trend is concerning, as the psychotic effects induced by cannabis is THC dose dependent [43, 44] and CBD has the potential to counteract THC-induced paranoia, memory impairment and positive psychotic symptoms and may provide some therapeutic benefit on its own [25, 45, 46].

Although consistent with other cannabinoid concentrations reported throughout the literature, no universal standards for laboratory testing exist and previous studies have

demonstrated the tendency for some laboratories to consistently report higher cannabinoid concentrations than their counterparts [28], which might contribute to the lack of accuracy in THC or CBD content in the labels of cannabis products [47]. Regardless of the method or laboratory used to measure THC content, the average THC concentration in all cases has been reported to be above 15% since 2014 [28]. Thus, the THC content marketed online is consistent with reports in which THC was actually measured, suggesting a high level of concordance of our data with the existing literature, and suggesting the degree of inaccuracy of our data is similar to these studies. In addition to discrepancies between laboratories, a recent study demonstrated that CBD products advertise an accurate CBD concentration only 31% of the time and unlabeled THC was detected in about 21% of products [47]. Such results suggest a need for regulations that improve transparency in regard to the location and methods by which each product was tested.

The inability to confirm the accuracy of reported THC and CBD is undoubtedly a limitation of this study. However, the concentrations reported in this study are representative of what a consumer would assess and therefore hold significant implications. The accuracy, or lack thereof, of labeled cannabinoid content of legal cannabis products poses a concern over patient's misconceptions regarding the potency of products they consume. Such misconceptions lead to either unintended adverse effects due to use of a higher potency product than anticipated, or to a perception of treatment failure due to use of a lower potency product than expected. Although they may not represent completely accurate cannabinoid concentrations, the cannabinoid content documented in our study are representative of what a patient would assess prior to selecting a product of cannabis for medical use. As mentioned before, online information is a potent tool to promote a favorable attitude towards high potency cannabis for medicinal purposes [7], namely for the treatment of chronic pain. However, our data suggests that medicinal programs in the U.S. are not using scientific evidence to develop a legal framework for the safe provision of medicinal cannabis. While these states do offer products amenable to medicinal use (THC <5–10%), these products are not what consumers primarily purchase. A Washington state cannabis potency and prices study demonstrated that flowers with THC > 15% accounted for over 90% of sales while flowers with THC <10% accounted for about 2% of expenditures between 2014 and 2016 [29]. This evidence suggests that having some products suitable for medicinal use is not sufficient to provide safe treatment to patients seeking help in medical cannabis dispensaries, perhaps in part due to online advertisement of primarily high potency products. We recognize that other factors could be at play in the selection of high potency cannabis in medicinal programs. For one, more potent products are better regarded in recreational cannabis practices [13], and this might influence the election of high THC products by recreational users [48]. Furthermore, since the price of a product increases with increasing THC concentration, revenue could contribute to budtender recommendations [29, 49]. In line with this assumption, a California study shows that over 40% of budtenders view medical decision-making as "less important" when making a recommendation [50]. In addition, 70% of all dispensary staff reported that a lack of knowledge served as a barrier to making a medical recommendation [50]. This trend is exacerbated by the fact that less than half of dispensaries across the country advise patients of potential side effects and even fewer warn of potential contraindications, while many promote ill-supported medical benefits [27].

The combination of these factors with our findings on online offering sets the grounds for policy makers and regulatory agencies to take action and guide towards the adoption of evidence based practices. Regulatory bodies have already adopted the practice of overseeing medications deemed safe for over the counter (OTC) use at low doses but with serious dose-dependent adverse effects. Such is the case with ibuprofen, which can be obtained OTC at the low doses used for mild to moderate pain but requires a prescription for the higher doses used

for rheumatoid arthritis and osteoarthritis. In 2014, the FDA concluded the need for a new OTC label that conveys the seriousness of this risk [51]. This is in stark contrast to the paucity of regulation regarding the labeling and advertising of the dose-dependent adverse risks of cannabis in legal programs in the U.S. Additionally, OTC medications that are considered safe but that convey the risk of abuse are often highly regulated, as is the case with pseudoephedrine. Interestingly, Uruguay regulates cannabis in a manner similar to how the U.S. regulates pseudoephedrine, requiring it be purchased in a pharmacy and limiting the amount that can be purchased in the course of a week and over a month [52]. Of note, the legal products offered in Uruguay have a THC concentration that is <10% to reduce the risk of dependence [53].

Limiting the availability of cannabis products for medical purposes to those with moderate THC concentrations (<10%) would reduce the risk of acute side effects and long-term undesirable outcomes. In line with this assumption, the availability of higher potency cannabis products has been linked to higher blood levels of THC in users that have been apprehended for driving under the influence of drugs [54]. In addition, it has been demonstrated under experimental conditions that when cannabis products of varying potency are made available to users who were asked to use sufficient amounts to achieve a given psychotropic state, the election of high potency products resulted in the exposure to higher amounts of THC [55]. It is logical and tempting to think that the amount of product that is consumed is more important than its potency and that cannabis users are likely to titrate the amount of product consumed based on its strength. Indeed, it has been shown that users of high potency cannabis inhale less volumes; however, they use larger amounts of cannabis and are therefore exposed to higher levels of THC compared to users of low potency products [56]. This shows that real titration does not exist when high potency products are used. These studies demonstrate that psychosocial variables are more relevant than pharmacological variables in cannabis use behavior [55, 56]. This, together with the fact that high THC exposure is a risk factor to develop cannabis use disorder and that cannabis smoking behavior is a predictive factor to the severity of cannabis dependence (regardless of THC exposure, [56]) warrants regulations that disallow high potency products for medicinal use. These concepts should be also considered within the recreational realm.

A necessary first step for new regulatory policies is a clear differentiation between products used in medical versus recreational cannabis programs. States with comprehensive medicinal marijuana programs do not often pass legislation limiting the strength of cannabis on the market [57]. Legislators should consider stricter regulations than are currently in place in order to deliver products or chemotypes more amenable for pain and provide a safer health care service. Regulations may include the addition of staff with proper training in pharmacology, patient counseling, and continual education in public health care. More collaboration among health professionals, scientists in pharmacology and pain specialists is needed to aid in the development of a more suitable legal framework for the provision and promotion of medicinal cannabis.

In conclusion, our findings suggest that medicinal programs are operating in a similar fashion to recreational programs based on the products they offer online (high THC), which are not adequate for medical use and could contribute to risky misconceptions towards medicinal cannabis. To combat this, states might consider collaboration with healthcare professionals to develop a more suitable legal framework for safe medicinal cannabis use across the United States, which could serve as a model to other countries when considering medical cannabis legalization.

## Supporting information

**S1 Fig. Frequency histograms of products in relation to THC per state.** Abundance (frequency) of products in relation of given THC contents (potencies). Note that X axis varies among states. Y axes are similar in all graphs, however inset graphs show smaller Y axes' scales to better represent data from states with less abundance products.
(TIFF)

**S2 Fig. Potency (THC content) histograms for individual products per state.** Every individual product offered in a given state is plotted on the X axis and its THC content is shown on the Y axis. Note that Washington state is presented with a larger X axis scale since this state offers many more products than the other surveyed states.
(TIFF)

**S3 Fig. Frequency histograms of products in relation to CBD per state.** Abundance (frequency) of products in relation of given CBD contents. Note that X axis varies among states. Y axes are similar in all graphs, however inset graphs show smaller Y axes' scales to better represent data from states with less abundance products.
(TIFF)

**S4 Fig. CBD content histograms for individual products per state.** Every individual product offered in a given state is plotted on the X axis and its CBD content is shown on the Y axis. Note that Washington state is presented with a larger X axis scale since this state offers many more products than the other surveyed states. In most states, the vast majority of products have 0% CBD, as shown on X axis towards the left.
(TIFF)

**S1 Table. Legalized United States recreational programs.**
(DOCX)

**S2 Table. Legalized United Stated Medical Programs.** [a] At time of data collection, cannabis recently legalized for medical purposes but did not have any active and open dispensaries. [b] The two dispensaries in Delaware exist under the same license. Only one website exists for both locations. [c] At time of data collection, there was a hold on Maryland cannabis practices. Therefore, no data was collected in Maryland.[d] Shortly after data collection, Massachusetts legalized cannabis for recreational purposes. For the purposes of this study, MA was considered a medical-only state.
(DOCX)

**S3 Table. Descriptive statistics for THC (%, top) and CBD concentrations (%, bottom) in all products offered in each sampled state.**
(DOCX)

**S4 Table. Comparisons of THC concentrations (%) in all products between each sampled state.** One way-ANOVA followed by Turkey's multiple comparisons test was used, and P values are reported. A $P<0.05$ was considered statistically significant. ns = not statistically significant.
(DOCX)

**S5 Table. Comparisons of CBD concentrations (%) in all products between each sampled state.** One way-ANOVA followed by Turkey's multiple comparisons test was used, and P values are reported. A $P<0.05$ was considered statistically significant. ns = not statistically

significant.
(DOCX)

**S6 Table. Descriptive statistics for THC concentrations (%) in all products offered in each sampled state separated by % THC categories (<5%, >5<10% THC, >10<15% THC, >15% THC).** ND = no data
(DOCX)

**S7 Table. Comparisons of THC concentrations (%) in all products between each sampled state separated by % THC categories (<5%, >5<10% THC, >10<15% THC, >15% THC).** One-way ANOVA followed by Turkey's multiple comparisons test was used, and P values are reported. A P<0.05 was considered statistically significant. ns = not statistically significant.
(DOCX)

**S8 Table. Descriptive statistics for CBD concentrations (%) in all products offered in each sampled state separated by % THC categories (<5%, >5<10% THC, >10<15% THC, >15% THC).** ND = no data
(DOCX)

**S9 Table. Descriptive statistics for THC (%, top) and CBD concentrations (%, bottom) in all products offered in CO and WA medical and recreational programs.**
(DOCX)

**S10 Table. Descriptive statistics for THC concentrations (%) in all products offered in CO and WA medical and recreational programs separated by % THC categories (<5%, >5<10% THC, >10<15% THC, >15% THC).**
(DOCX)

**S11 Table. Descriptive statistics for CBD concentrations (%) in all products offered in CO and WA medical and recreational programs separated by % THC categories (<5%, >5<10% THC, >10<15% THC, >15% THC).**
(DOCX)

**S1 Data.**
(XLSX)

## Acknowledgments

The authors would like to acknowledge the Department of Anesthesiology at Wake Forest University School of Medicine for funding.

## Author Contributions

**Conceptualization:** E. Alfonso Romero-Sandoval.

**Data curation:** Mary Catherine Cash, Katharine Cunnane, Chuyin Fan, E. Alfonso Romero-Sandoval.

**Formal analysis:** Mary Catherine Cash, Katharine Cunnane, E. Alfonso Romero-Sandoval.

**Funding acquisition:** E. Alfonso Romero-Sandoval.

**Investigation:** Mary Catherine Cash, Katharine Cunnane, Chuyin Fan, E. Alfonso Romero-Sandoval.

**Methodology:** E. Alfonso Romero-Sandoval.

**Project administration:** E. Alfonso Romero-Sandoval.

**Resources:** E. Alfonso Romero-Sandoval.

**Supervision:** E. Alfonso Romero-Sandoval.

**Writing – original draft:** Mary Catherine Cash, Katharine Cunnane, E. Alfonso Romero-Sandoval.

**Writing – review & editing:** Mary Catherine Cash, Katharine Cunnane, Chuyin Fan, E. Alfonso Romero-Sandoval.

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
