## [Decision Letter · Decision Letter 0]

25 Oct 2019

PONE-D-19-25611

Mapping cannabis potency in medical and recreational programs in the United States.

PLOS ONE

Dear Dr. Romero-Sandoval,

Thank you for submitting your manuscript to PLOS ONE. After careful consideration, we feel that it has merit but does not fully meet PLOS ONE’s publication criteria as it currently stands. Therefore, we invite you to submit a revised version of the manuscript that addresses the points raised during the review process.

ACADEMIC EDITOR: The manuscript is of high interest and importance to the field.  Three independent referees evaluated the manuscript with the majority recommending major revisions. It is recommended that the authors read the comments provided by the reviewers and address completely.

We would appreciate receiving your revised manuscript by November 15, 2019. To enhance the reproducibility of your results, we recommend that if applicable you deposit your laboratory protocols in protocols.io, where a protocol can be assigned its own identifier (DOI) such that it can be cited independently in the future. For instructions see: http://journals.plos.org/plosone/s/submission-guidelines#loc-laboratory-protocols

We look forward to receiving your revised manuscript.

Kind regards,

Tally M. Largent-Milnes

Academic Editor

PLOS ONE

Journal Requirements:

2) Please include in your methods section, the methods used to identify licensed cannabis dispensaries in the different states.

3) Please include captions for your Supporting Information files at the end of your manuscript, and update any in-text citations to match accordingly. Please see our Supporting Information guidelines for more information: http://journals.plos.org/plosone/s/supporting-information.

Reviewers' comments:

Reviewer's Responses to Questions

**Comments to the Author**

1. Is the manuscript technically sound, and do the data support the conclusions?

Reviewer #1: Partly

Reviewer #2: Partly

Reviewer #3: Yes

2. Has the statistical analysis been performed appropriately and rigorously? 

Reviewer #1: Yes

Reviewer #2: Yes

Reviewer #3: Yes

3. Have the authors made all data underlying the findings in their manuscript fully available?

Reviewer #1: Yes

Reviewer #2: Yes

Reviewer #3: Yes

4. Is the manuscript presented in an intelligible fashion and written in standard English?

Reviewer #1: Yes

Reviewer #2: Yes

Reviewer #3: Yes

5. Review Comments to the Author

Reviewer #1: This manuscript surveys and compares the reported THC and CBD concentrations found in cannabis distributed for medicinal versus recreational purposes across many US states. This is an extensive study accessing publicly available information from websites maintained by the cannabis distributors. They found that THC and CBD concentrations are generally very high and very variable. Recreational cannabis has slightly higher concentrations of THC and CBD with large state to state variability. This information is potentially important for the medical and scientific communities, however, the conclusions by the authors that access to low (and in their opinion more effective) concentration THC from medical distributors are an over interpretation of the findings. Further, determining actual concentrations of THC and CBD, while very difficult at this scale, and comparing to the reported values would be more enlightening. Below are specific comments on the manuscript:

1. In the 2nd paragraph of the introduction, it is stated that an increase in THC content results in a loss of analgesic effects. However, the cited studies actually report superior analgesic effects from “high” THC content of ~7% THC products than from “low” or “medium” THC content of 1-4% THC products, as well as impaired performance on certain tests such as the completion time of the Paced Auditory Serial Addition test and Trail Making test. Further, these “high” THC product concentrations are in the middle range of the concentrations suggested to be efficacious for pain with minimal psychotropic effects in the next line. These statements and citations are inconsistent with respect to analgesic efficacy and adverse events.

2. Much more important than the concentration of cannabinoids in plant material, and not mentioned here or elsewhere, is the quantity of plant material per administration. The potency of plant material is far less important than determining the quantity of cannabinoids administered.

3. For the potency of cannabis products to be meaningful, the quantity of cannabis products consumed must also be known to determine the total cannabinoids administered. For high potency cannabis products to pose a greater health risk, they would have to be consumed in equal quantities as lower potency products, but no evidence of this is presented and this important qualification is never stated. Statements claiming that the potency of cannabis products per se pose a health risk should be removed or qualified to address this.

4. It is stated that data was collected only for cannabis flower and pre-rolls. However, the range of the reported data includes strains with approximately 90% THC by weight, or over 50% CBD by weight. These outliers are certainly concentrated cannabis extracts or formulations, not flower. This should be addressed and the data checked for accuracy.

5. At the end of paragraph 4 in the discussion, it is mentioned that strains higher in THC were lower in CBD. This inverse relationship is a reported result of selection for either cannabinoid, and the increase in high CBD and low THC cannabis strains is not mentioned here.

Reviewer #2: The current manuscript entitled “Mapping cannabis potency in medical and recreational programs in the United States” addresses a very important issue in medical cannabis use in the US. An accretion of evidence supports the public health risks of the regular use of high-THC products; patients, healthcare providers, and policy makers would all benefit from enhanced attention to the risks involved with potent THC products. However, in order for the manuscript to be suitable for publication in PLOS, it needs significant revisions to the Introduction, Results, Figures, and Discussion.

Major concerns:

1) The foundation of the manuscript is based on the assumption that the medically-acceptable phenotype of analgesic cannabis is a flower with 5-10% THC. Although the analgesic efficacy of low-potency flower has been demonstrated in the literature (which the authors properly cite), there is an over-reliance on this single study throughout the manuscript, and a failure to acknowledge the large body of evidence which also shows the analgesic efficacy of unknown cannabis phenotypes, as well as recent studies showing that THC potency is the strongest predictor of analgesia (Le et al. 2019, PMID 31519268): in other words, the exact opposite of the authors’ presumed medically-relevant phenotype. There is not yet enough high-quality evidence for the scientific/medical communities to agree upon what a medical cannabis phenotype is (that which has maximal efficacy and minimal side effects).

That said, there is ample evidence to support the RISKS associated with THC potency: this is the stronger argument for the evidence presented within the manuscript. The introduction and conclusion should clearly state for the reader that there is likely a very wide therapeutic window (in terms of analgesia) for THC, but that the risks of chronic consumption at the upper limit of the therapeutic window may be detrimental (cannabis use disorder, hyperemesis, cognitive impairment etc.).

2) Although the dataset is thorough and sufficient for analysis, extreme outlying values in THC potency warrant the inclusion of the median, in addition to the mean THC potencies found in each legal market. Furthermore, the presentation of the data in the figures is not particularly intuitive. Some of the figures are difficult to read, either because of overcrowding with unnecessary statistical comparisons, or low-resolution of images. The inclusion of potency histograms would be of great utility to the readership of PLOS. Also, in the Introduction the authors mention the importance of the ratio of CBD:THC, however this is never directly presented in the figures.

Minor comments:

There are several instances of inaccurate interpretations of the cited literature throughout the manuscript. There is also a heavy reliance upon meta-analyses, commentaries, and systematic reviews instead of primary research articles. The authors utilize colloquial language throughout the manuscript, which strongly warrants correction: the readers of PLOS are looking to the journal to provide the most up-to-date and scientifically accurate information about cannabis, which should not include the words “strain” “indica” or “sativa,” unless specifically discussing the mythology and scientific inaccuracies that these terms impart. Furthermore, the terms psychotropic and psychoactive warrant clear differentiation, as do the psychological phenomenon of euphoria and impairment.

By addressing these comments and providing a more detailed analysis of the data (median), and presenting this data in more intuitive figures, the readers of PLOS would undoubtedly benefit from the evidence the authors present here.

Reviewer #3: This manuscript reports the results of a cannabis potency surveyed done using dispensaries websites and how it relates to their medical and recreational use. It is well accepted that recreational cannabis-based produces are driven by high content in THC, whereas medical products contain lower amounts of THC and are often supplemented with CBD. The study is well designed, and the interpretation of the data is thorough. Highlighting the misconceptions toward cannabis-based products for recreational versus medical use is timely, a relevant question and will help reduce the risks of pain patients in using products that contain recreational amounts of THC.

Limitation of the study: the manuscript clearly mentions that the information collected was from dispensaries online and discusses specifies limitations. One aspect that should be emphasized more is that, at least in WA state, the values of THC and CBD on cannabis product labels are not always accurate. Another limitation that should be mentioned is that some states allow patients to grow their medical cannabis and thus this population is not included in this study.

“A total of 8,505 cannabis strains across 653 dispensaries were sampled”. The specifics of what constitutes a “strain” is needs to be clarified. This reviewer was not aware that there were so many strains. Adding a reference here would help the reader better understand the extent of the diversity in strains.

“a decline in the number of opioid prescriptions”. Several recent studies show that this is not always the case, depending on how the results are analyzed and whether recent years data are included or not. These studies concluding the opposite view, that legalization has not affected opioid use, need to be mentioned here.

“accessible in a non-uniformly regulated market in contrast to other countries”. This reads as a bold statement. More details will help the reader better understand the context of this statement.

“Alternatively, CBD ameliorates the euphoric effects and counteracts the unintended adverse effects of THC (14)”. This sentence is confusing. How can CBD ameliorate the euphoria of something that is already euphoric, and what are the side effects of THC that are referred to here?

“This study will also evaluate the appropriateness of available strains for medicinal use”. What level of diligence is dedicated to validating the strains? Where there any genetic profiling done? Some language should be added here in the manuscript.

The resolution of the various figures and graphs was low in this reviewers PDF file and should be improved.

6. PLOS authors have the option to publish the peer review history of their article (what does this mean?). If published, this will include your full peer review and any attached files.

Reviewer #1: No

Reviewer #2: Yes: Adrianne Wilson-Poe

Reviewer #3: No

---

## [Author Response · Author response to Decision Letter 0]

27 Dec 2019

A point by point response to reviewers document is included in the uploaded documents.

---

## [Decision Letter · Decision Letter 1]

25 Feb 2020

Mapping cannabis potency in medical and recreational programs in the United States

PONE-D-19-25611R1

Dear Dr. Romero-Sandoval,

We are pleased to inform you that your manuscript has been judged scientifically suitable for publication and will be formally accepted for publication once it complies with all outstanding technical requirements.

With kind regards,

Tally Largent-Milnes

Academic Editor

PLOS ONE

Additional Editor Comments (optional):

Reviewers' comments:

Reviewer's Responses to Questions

**Comments to the Author**

1. If the authors have adequately addressed your comments raised in a previous round of review and you feel that this manuscript is now acceptable for publication, you may indicate that here to bypass the “Comments to the Author” section, enter your conflict of interest statement in the “Confidential to Editor” section, and submit your "Accept" recommendation.

Reviewer #1: All comments have been addressed

Reviewer #2: All comments have been addressed

2. Is the manuscript technically sound, and do the data support the conclusions?

Reviewer #1: Yes

Reviewer #2: Yes

3. Has the statistical analysis been performed appropriately and rigorously? 

Reviewer #1: Yes

Reviewer #2: Yes

4. Have the authors made all data underlying the findings in their manuscript fully available?

Reviewer #1: Yes

Reviewer #2: Yes

5. Is the manuscript presented in an intelligible fashion and written in standard English?

Reviewer #1: Yes

Reviewer #2: Yes

6. Review Comments to the Author

Reviewer #1: (No Response)

Reviewer #2: The authors have sufficiently addressed all major concerns that I had with the original manuscript. The quality of the figures is much improved, legibility and clarity are sufficient for publication. Data outliers have been removed such that the analysis is more accurate. Vernacular has been clarified throughout the manuscript.

7. PLOS authors have the option to publish the peer review history of their article (what does this mean?). If published, this will include your full peer review and any attached files.

Reviewer #1: No

Reviewer #2: Yes: Adrianne Wilson-Poe

---

## [Editor Report · Acceptance letter]

3 Mar 2020

PONE-D-19-25611R1 

Mapping cannabis potency in medical and recreational programs in the United States 

Dear Dr. Romero-Sandoval:

I am pleased to inform you that your manuscript has been deemed suitable for publication in PLOS ONE. Congratulations! Your manuscript is now with our production department. 

With kind regards,

on behalf of

Dr. Tally Largent-Milnes 

Academic Editor

PLOS ONE